# Exploring the Efficacy of a Virtual First Year Interprofessional Education Event

**DOI:** 10.3390/healthcare10081539

**Published:** 2022-08-14

**Authors:** Isdore Chola Shamputa, Boon Kek, Loretta Waycott, Tammie Fournier, Shaun McCarville, John Doucet, Derek J. Gaudet, Marc Nicholson

**Affiliations:** 1Department of Nursing & Health Sciences, University of New Brunswick, Saint John, NB E2L 4L5, Canada; 2Dalhousie Medicine New Brunswick, Saint John, NB E2K 5E2, Canada; 3Allied Health Department, New Brunswick Community College, Saint John, NB E2J 4C5, Canada; 4Department of Psychology, University of New Brunswick, Saint John, NB E2L 4L5, Canada

**Keywords:** interprofessional education, virtual activity, interprofessional collaborative competency attainment survey

## Abstract

Interprofessional education (IPE) activities are utilized in health education programs to develop interprofessional collaboration (IPC) competencies. All first-year healthcare students at three postsecondary learning institutions attend a mandatory introductory IPE event annually. During the 2020/2021 academic year, the event was moved from a face-to-face activity to a virtual format due to the COVID-19 pandemic restrictions. This study examined whether the virtual IPE activity was effective in supporting the development of interprofessional competencies for first-year healthcare students. Two hundred and six students attended a synchronous didactic presentation on IPE competencies and discussed a simulated case in interprofessional groups of eight students and two faculty facilitators. The Interprofessional Collaborative Competency Attainment Survey (ICCAS) was used to measure the students’ opinions on interprofessional competencies. Paired *t*-tests were used to compare the pre- and post-scores. One hundred and nine (52.9% response rate) students completed the survey. Surveys from 99 students with matched pre- and post-scores were included in the study. The ICCAS competencies showed improvements (*p* < 0.05) in all of the students’ self-reported IPE competencies following the activity compared to before the training. Our findings indicate that the virtual IPE activity is effective in facilitating the development of IPC for first-year healthcare students.

## 1. Introduction

Interprofessional education (IPE) activities are an important part of the curriculum for healthcare programs and are utilized to develop interprofessional collaboration (IPC) competencies. IPE involves at least two students from different health disciplines learning about, from, and with each other to support collaboration and improve patient health outcomes [1]. The Canadian Interprofessional Health Collaborative (CIHC) has identified six core domains necessary for IPC, which are role clarification, team functioning, interprofessional communication, patient/family/community-centered care, interprofessional conflict resolution, and collaborative leadership [2]. Until recently, IPE activities were typically conducted using the traditional face-to-face format, as communication and interaction were key components of the subject matter. With the advent of the coronavirus disease 2019 (COVID-19) pandemic and subsequent public health restrictions including lockdowns, many learning institutions had to shift to new teaching modalities. The switch to online teaching presented the option to conduct virtual IPE activities to ensure that curriculum needs were still met. However, such a modality had not been conducted for IPE activities previously at the University of New Brunswick (UNB), New Brunswick Community College (NBCC), and Dalhousie Medicine New Brunswick (DMNB), collectively referred to as Tucker Park campus.

Consequently, the purpose of this study was to assess whether a virtual IPE activity was effective in supporting the development of interprofessional competency for first-year healthcare students from different programs and institutions.

A literature search at that time revealed a dearth of available published research on this topic. Some studies had indicated that virtual online sessions may increase student confidence in clinical decision making [3], increase students’ knowledge of other professional roles [4] and increase a student’s perception of team performance [5]. Since then, several studies have demonstrated the effectiveness of virtual learning with students achieving learning outcomes similar to in-person experiences [6,7], reporting online learning as a positive experience [8,9] and findings of higher student performance in online learning strategies compared to in-person learning, with no difference in student satisfaction [10].

### Background

“Tuckerpalooza” is a mandatory introductory IPE activity for all first-year students enrolled in health profession programs on the Tucker Park campus. The event is designed to introduce IPC concepts and patient-centered care to the students as they embark on their education and training. This annual event was held virtually for the first time during the 2020/2021 academic year because of the public health restrictions on face-to-face activities due to the COVID-19 pandemic. The event was held on 29 September 2020 for 1 h 30 min via Microsoft (MS) Teams. “Tuckerpalooza” was divided into two parts. During the first part of the event, all first-year students from the three participating institutions joined a synchronous MS Teams session conducted in a lecture-style format. During this portion, faculty representatives from the respective programs across the three institutions presented IPE concepts to the students and a short introduction to the different healthcare programs present. Finally, there was a presentation of a fictional IPE-related case study on the recent COVID-19 pandemic. Prior to dispersing the students and facilitators to their discussion groups, they were informed of the background and goals of the study.

Thereafter, the students logged on to separated MS Teams meeting channels, which served as breakout rooms in their predetermined interprofessional groups of eight students to discuss the presented case study. These groups contained no more than two participants from the same professional discipline. Two faculty from different disciplines who had received facilitator training facilitated the discussions in each breakout room with their assigned group of interprofessional students. After that, the case study was employed to help guide the discussion on IPE/IPC with students. The facilitators were provided with guiding questions to focus the discussion. The key topics were on role clarifications, collaborative patient-family-centered approach, and resource allocation while using the pandemic as a backdrop. Shortly after the breakout room discussions, students were invited to complete an evaluation survey online voluntarily. The web links to the informed consent and evaluation tool were provided to the students via email at the conclusion of the event.

## 2. Materials and Methods

### 2.1. Design

This study used a retrospective pretest–posttest design, administered simultaneously after the IPE intervention. To determine the effectiveness of the virtual intervention, students’ ICCAS pre-scores were compared to their post-scores.

### 2.2. Setting

This study was conducted on the Tucker Park campus in Saint John, New Brunswick, Canada, with the students and facilitators participating virtually.

### 2.3. Study Population

The study population comprised 206 first-year healthcare students who attended an introductory IPE event on 29 September 2020. Although participation in the study was optional, the event was a mandatory part of their curriculum. They included Bachelor of Nursing (52), Medicine (31); Practical Nurse (34); Medical Laboratory Science (26); Personal Support Worker (25); Pharmacy Technician (11); Respiratory Therapy (17) and Radiological Technology (10) programs. Some Radiological Technology students were from a distributed campus in another city.

### 2.4. Recruitment and Data Collection

#### Instrument

Data were collected using the Interprofessional Collaborative Competencies Attainment Survey (ICCAS) [11]. The ICCAS is an instrument intended to measure the self-reported change in interprofessional competency [12]. The tool contains 20 statements to address six competencies related to interprofessional care communication (five statements), collaboration (three statements), roles and responsibilities (four statements), collaborative patient-family-centered approach, and conflict management/resolution (three statements each), and team functioning (two statements). Participants rated their competencies on a seven-point Likert-type scale where 1 = strongly disagree; 2 = moderately disagree; 3 = slightly disagree; 4 = neutral; 5= slightly agree; 6 = moderately agree; 7 = strongly agree; and na = not applicable. Completion of the ICCAS was anonymous and voluntary.

### 2.5. Data Analysis

Data analysis was conducted using Microsoft Excel and the IBM SPSS software, version 27 (Armonk, NY, USA). The response category “not applicable” was considered a missing value in the data analysis. The mean (M) and standard deviation (SD) for each item is presented along with the pre- and post-ICCAS scores. The mean (total) score of all the statements was also computed to provide a global assessment of overall improvement. Two-tailed paired *t*-tests were used to compare individuals’ scores on pre- and post-test ICCAS surveys. A *p*-value of <0.05 was considered statistically significant. The ICCAS scale(s) internal consistency was assessed using Cronbach’s alpha (α). In this study, the ICCAS demonstrated a very high internal consistency (α = 0.97). Cohen’s *d*, which indicates the number of standard deviations by which the pre- and post-tests differ, was employed to assess the effect size. Cohen’s *d* values between 0.2 and 0.5 are considered a ‘small’ effect size, values between 0.5 and 0.8 represent a ‘medium’ effect size, and values greater than 0.8 are interpreted as a ‘large’ effect size. A Cohen’s *d* value of less than 0.2 indicates a negligible difference [13].

## 3. Results

A total of 206 first-year students enrolled in eight healthcare programs from three post-secondary learning institutions attended an introductory IPE event. Responses were received from 109 students (52.9%), of whom 10 were excluded because of missing data. Responses from the remaining 99 students were included in the study. The results of two-tailed paired *t*-tests before and after the IPE event for the six competencies and all of the 20 individual statements are presented in Table 1.

The mean summed score of all test statements improved from 5.34 (SD = 1.11) before the IPE event to 6.14 (SD = 0.95) after the IPE event (*p* < 0.001) (Table 1). Statistically significant change was noted for all six competencies and the 20 individual statements within each domain (*p* < 0.001), indicating an increase in scores on each statement from pre- to post competency scores. Moderate to large pre-/post-IPE event effect sizes were observed for 19 of the 20 statements in the ICCAS. The largest effect size was observed in three statements: (1) seek out IP members to address the issue (statement 6), (2) understand the abilities and contributions to the IP team (statement 11), and (3) negotiate responsibilities within overlapping with IP team members (statement 20). Medium size effects were observed in most of the remaining statements except one; express my ideas and concerns without being judgmental, where the only “small” effect size was observed (statement 3) (Table 1).

## 4. Discussion

This study aimed to assess whether a virtual IPE activity was effective in supporting the development of interprofessional competency for first-year healthcare students at three post-secondary institutions in New Brunswick. The results showed a statistically significant improvement in all of the students’ self-perceived competencies in all domains of the ICCAS following participation in the event, indicating that the virtual IPE event was effective in developing interprofessional competency in healthcare students.

Our results are consistent with previous findings where significant differences were achieved in all 20 ICCAS statements [14,15,16]. However, while the response rate was lower in our study (52.9%), as was the case with one study (60%) [15], it was much lower than that reported by two other studies (95.5% and 96.5%) [14,15]. This variance could partly be attributed to the mandatory participation in one study [16] compared to the other studies, although participation in the study with the highest response rate was also voluntary [14]. More recent reports using qualitative, quantitative, and mixed methods have also demonstrated the effectiveness of virtual IPE curriculum activities where students have improved in their interprofessional understanding [7], increased their understanding of professional roles and team collaboration [8,9], and strengthened their understanding of the value of effective communication for patient care [9].

The high internal consistency reported in this study is in accord with prior studies [12,17,18,19], further attesting to the suitability of the ICCAS for assessing IPC competencies.

A side benefit of the successful virtual Tuckerpalooza was that it provided an opportunity for some students in at least one program (Radiological Technology) on a distributed campus in another city to participate in the event for the first time. This suggests that virtual IPE can be used to potentially expand the program to students in healthcare programs at other universities in the region with limited opportunities for IPE.

Studies on virtual IPE hitherto suggested that this mode of teaching is effective in transferring the IPE knowledge to healthcare students and should be widely utilized by post-secondary learning institutions to help meet program objectives.

This study has some limitations; the low response rate in this study might lend its findings to nonresponse bias. We were unable to determine whether responses from one professional discipline were over- or under-represented because participation was anonymous; this was due to responses being tabulated as a whole versus per profession. However, it is worth noting that the ICCAS has been widely validated and is the frequently preferred tool for assessing students’ self-perceived attainment of IP collaborative competencies. Future virtual IPE activities should be considered to confirm findings reported in this study and report findings by professional discipline.

## 5. Conclusions

This study contributes to the IPE/IPC field by demonstrating that the virtual “Tuckerpalooza”, previously offered only in person, is suitable in facilitating the development of IPC competencies in healthcare students. Additionally, the virtual medium’s effectiveness presents institutions with additional opportunities for IPE collaboration to allow students to gain more exposure and understanding of different healthcare professions post-pandemic.

## Figures and Tables

**Table 1 healthcare-10-01539-t001:** Self-Perceived Interprofessional Collaborative Competency Attainment Survey (ICCAS) of First-Year Healthcare Students Before and After Participating in an Interprofessional Education Event (N = 99).

ICCAS Competencies	Individual Statements	Pre-IPE Event Mean ^a^ (SD)	Post-IPE Event Mean ^a^ (SD)	*t*-Value	*p* Value ^b^	Cohen’s *d*	Qualitative Difference ^c^
Communication	Mean score	5.54 (1.18)	6.18 (0.98)	7.34	<0.001	0.74	Medium
1	Promote effective communication among members of an interprofessional (IP) team	5.24 (1.43)	6.06 (1.16)	7.05	<0.001	0.71	Medium
2	Actively listen to IP team members’ ideas and concerns	5.96 (1.24)	6.49 (0.96)	5.60	<0.001	0.56	Medium
3	Express my ideas and concerns without being judgmental	5.81 (1.32)	6.24 (1.09)	4.55	<0.001	0.46	Small
4	Provide constructive feedback to IP members	5.16 (1.41)	5.99 (1.14)	7.38	<0.001	0.74	Medium
5	Express my ideas and concerns in a clear, concise manner	5.52 (1.40)	6.13 (1.09)	5.38	<0.001	0.54	Medium
Collaboration	Mean score	5.21 (1.26)	6.06 (1.00)	8.85	<0.001	0.89	High
6	Seek out IP members to address the issue	4.92 (1.38)	5.96 (1.08)	9.14	<0.001	0.92	High
7	Work effectively with IP members to address the issue	5.37 (1.29)	6.08 (1.05)	6.81	<0.001	0.69	Medium
8	Learn with, from and about IP team members to enhance care	5.33 (1.38)	6.15 (1.00)	7.52	< 0.001	0.76	Medium
Roles and Responsibilities	Mean score	5.24 (1.23)	6.16 (1.02)	8.89	<0.001	0.89	High
9	Identify and describe my abilities and contributions to the IP team	5.17 (1.21)	6.05 (1.07)	7.95	<0.001	0.80	Medium
10	Be accountable for my contributions to the IP team	5.51 (1.33)	6.20 (1.02)	6.46	<0.001	0.65	Medium
11	Understand the abilities and contributions to the IP team	5.11 (1.37)	6.18 (1.13)	8.77	<0.001	0.88	High
12	Recognize how others’ skills and knowledge complement and overlap with my own	5.18 (1.50)	6.20 (1.15)	7.49	<0.001	0.75	Medium
Patient-Centered Approach	Mean score	5.19 (1.26)	6.06 (0.99)	7.71	<0.001	0.77	Medium
13	Use an IP team approach with the patient to assess the health situation	5.02 (1.43)	5.93 (1.07)	6.95	<0.001	0.70	Medium
14	Use an IP team approach with the patient to provide whole person care	5.08 (1.37)	6.03 (1.05)	7.38	<0.001	0.74	Medium
15	Include the patient/family in decision making	5.45 (1.21)	6.22 (1.04)	7.65	<0.001	0.77	Medium
Conflict Management	Mean score	5.64 (1.21)	6.35 (1.02)	7.44	<0.001	0.75	Medium
16	Actively listen to the perspectives of the IP team members	5.73 (1.24)	6.37 (1.07)	6.41	<0.001	0.64	Medium
17	Take into account the perspectives of IP team members	5.67 (1.30)	6.41 (1.02)	6.94	<0.001	0.70	Medium
18	Address team conflict in a respectful manner	5.53 (1.34)	6.26 (1.15)	7.22	<0.001	0.73	Medium
Team Functioning	Mean score	4.97 (1.37)	5.88 (1.25)	7.99	<0.001	0.80	High
19	Develop an effective care plan with IP team members	4.97 (1.40)	5.82 (1.31)	7.37	<0.001	0.74	Medium
20	Negotiate responsibilities within overlapping with IP team members	4.97 (1.40)	5.95 (1.24)	8.04	<0.001	0.81	High
Total	Mean score	5.34 (1.11)	6.14 (0.95)	9.15	<0.001	0.92	High

^a^ Scores were measured on a scale of 1 to 7, where 1 was strongly disagree and 7 was strongly agree; ^b^ Paired *t*-tests results of pre- and post-IPE event. Statistical significance was defined as *p* < 0.05; ^c^ Qualitative difference (Cohen’s *d* interpretation): *d* between 0.2 and 0.5 is considered small; *d* between 0.5 and 0.8 is considered medium; and *d* > 0.8 is considered large.

## Data Availability

The data presented in this study are available on request from the corresponding author.

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
