# Peer review of "Exploring the Efficacy of a Virtual First Year Interprofessional Education Event"

_healthcare, 2022, doi:10.3390/healthcare10081539_

Round 1
Reviewer 1 Report
Dear collegues
the manuscript is interesting. I suggest the following revisions
(a) the background section must be reviewed: first attention must be paid to the literature also with reference to the available evidence on distance learning (see for example Rossettini). Then the initiative and the activities carried out in your context.
(b) the method part is well constructed: however, there is no guideline (for example STROBE) that could increase the rigor in reporting.
(c) the ethics section needs to be strengthened. students are vulnerable and even with respect to low participation, this section must be implemented.
(d) the discussion should also be strengthened by comparing what emerged with the available evidence.
The study is interesting and I encourage the revisions mentioned above.
Reviewer 2 Report
In this delightful short paper, the authors report significant success at increasing inter-professional competencies using their one-time and remote IPE experience. This is good news for institutions that cannot form teams of healthcare students from various disciplines owing to the presence of only one or two healthcare programs at their own institutions. I have only two major suggestions.
1. The originators of the ICCAS survey (the authors’ references 7 and 13) maintain that “all ICCAS items load onto a single factor, (so) inter-professional care competencies are very interrelated.” The authors should include this finding in their paper and determine whether they obtained the same result.
2. In line 119 the authors state that they collected data on the students’ program of study, so they should be able to report results for each program as well as overall results.
A minor point: in lines 127-128, is the Cronbach’s alpha of 0.97 for the authors’ data or data from prior studies?
Author Response
Please see the attachement

Round 2
Reviewer 1 Report
Thank you for having considered my suggestions.
Reviewer 2 Report
The authors have adequately addressed each of my suggestions.